# Barriers of Occupational Safety Implementation in Infrastructure Projects: Gaza Strip Case

**DOI:** 10.3390/ijerph18073553

**Published:** 2021-03-29

**Authors:** Yazan Issa Abu Aisheh, Bassam A. Tayeh, Wesam Salah Alaloul, Amro Fareed Jouda

**Affiliations:** 1Civil Engineering Department, Middle East University, Amman 11831, Jordan; yabuaisheh@meu.edu.jo; 2Civil Engineering Department, Faculty of Engineering, Islamic University of Gaza, Gaza PO Box 108, Palestine; amro.f.jouda@gmail.com; 3Department of Civil and Environmental Engineering, Universiti Teknologi PETRONAS, Bandar Seri Iskandar, Tronoh 32610, Malaysia; wesam.alaloul@utp.edu.my

**Keywords:** barriers, infrastructure projects, health and safety, Gaza Strip

## Abstract

Infrastructure projects are the foundation for essential public services and have an influential position in societal development. Although the role of infrastructure projects is substantial, they can involve complexities and safety issues that lead to an unsafe environment, and which impacts the project key stakeholders. Therefore, this study aimed to evaluate the barriers to implementing occupational safety in infrastructure projects in the Gaza Strip, which cause serious threats and reduce project performance. To evaluate the barriers, 39 items were highlighted and modified as per the construction context and environment, and which later were distributed in the form of a questionnaire, to get feedback from consultants and contractors. The analysis shows that in the safety policy barriers group, consultants and contractors both ranked the item “a contractor committed to an occupational safety program is not rewarded” first. In the management barriers group, consultants and contractors both ranked the item “safety engineer does not have significant powers, such as stopping work when needed” in the first place. In the behavior and culture barriers group, consultants and contractors both ranked the item “workers who are not committed to occupational safety are not excluded” in the first place. Overall, both consultants and contractors shared the same viewpoint in classifying the barriers in the working environment. The outcome of this study is beneficial for Palestinian construction industry policymakers, so they can monitor the highlighted barriers in on-going infrastructure projects and can modify the safety guidelines accordingly.

## 1. Introduction

The construction industry contributes a valuable amount to the economy of any country, however, it is still badly affected by several issues such as cost overrun [1,2], time overrun [3,4,5], price fluctuation due to inflation [6,7,8], and most importantly due to poor safety performance [9]. Despite these severe issues, a project gets completed, but is not declared successful because the project failed to achieve the projects’ objectives [10,11]. Neglecting safety measures can result both in physical damage to workers and financial losses, which can increase the direct and indirect cost of the project [12,13]. As per the International Labour Organization (ILO), 2.3 million workers die each year worldwide due to occupational accident, impacting 4% of world GDP [14]. To successfully implement safety guidelines within a company, an equal effort is required by all the stakeholders of the construction industry [15]. In developed countries, considerable attention is provided to health and safety [16]. Whereas, in developing or underdeveloped countries the condition varies as the consideration of health and safety is far behind in its implementation [17,18,19]. The scenario can be different if collaborative work for occupational health and safety is performed between developed and developing countries, where sharing of information is done to bring major reforms [20,21]. 

One of the most important functions of infrastructure projects is to develop public services and improve the daily situation, necessary to support economic and social activity. In Palestine, infrastructure projects differ from the rest of the world due to the scarcity of financial resources and the density of population distribution. Infrastructure projects require continuous management and experience of the long-term planning and development of projects in line with plans. Infrastructure projects require substantial financial investment, time commitment, and significant human and material resources. Infrastructure development in Palestine is affected by many obstacles and problems that hinder current projects and plans, and the implementation of basic and public services [22]. One of the biggest challenges is the Israeli occupation, which controls the arrival of basic construction materials (cement, steel, and aggregates) to the Gaza Strip through long-term coordination mechanisms, requiring complex and time-consuming processes, which hinder current and future infrastructure projects [23]. Infrastructure projects in the Gaza Strip have been severely affected by lack of construction materials, in addition to the destruction incurred during military operations.

The Gaza Strip is the south western part of Palestine, which is located on the south-eastern coast of the Mediterranean Sea. The area of the Gaza Strip is about 365 km^2^, with an approximate length of 45 km, and a width between 6 and 12 km. It is confined between the Mediterranean Sea in the west, Egypt in the south, and occupied Palestine since 1948 in the east and the north, as shown in Figure 1. The Gaza Strip is located between longitudes 34°2″ and 34°25″ E, and latitudes 31°16″ and 31°45″ N [24]. 

The lack of planning and poor management in the selection and implementation of infrastructure projects suitable to current and future conditions are the problems that the Gaza Strip suffers, as there are no clear and effective criteria in determining the priorities for infrastructure projects. No priorities and long-term plans are explained for the stages of implementation of infrastructure projects [22]. According to the last report of the Palestinian Contractors Union [25], there were 294 classified contractors in the Gaza Strip in 2019. The construction activities are divided into five categories (roads, buildings, electromechanical, water/sewage, public works, and maintenance). Each one of these categories has five classes (from one to five). The contractor can have more than one classification in different categories of the above-mentioned five categories. The contractors of the first, second, and third classes for infrastructure works totaled 122 in the Gaza Strip in 2019.

The Gaza Strip suffers from a fragile infrastructure, however, there is growing interest in infrastructure development in various large projects, ranging from road projects, sewage plants, water desalination, electricity and others. This development should be accompanied by parallel attention to the requirements of occupational safety, especially since infrastructure projects have special characteristics among construction projects, as they relate to the public, whether in the stages of construction, maintenance, or demolition. Keeping to the discussion in consideration, a research question was established: “What are the barriers that hinder the occupational safety implementation in infrastructure projects in Gaza Strip”? Therefore, this research aims to assess the barriers to implementing occupational safety in infrastructure projects in the Gaza Strip from the consultants’ and contractors’ point of view, and with the set hypothesis: “The barriers affect the implementation of safety practices in the infrastructure sector in the Gaza Strip”.

## 2. Literature Review 

In infrastructure projects, factors and conditions surrounding construction sites vary significantly compared with other projects; the workers must perform dangerous jobs that require high physical and mental strength for long periods. In addition to unsanitary conditions, unavailable services, messy work environments, and exposure to bad weather conditions, the impression of infrastructure construction sites is that they are disorganized and dangerous [26]. Delivering projects on time requires effective planning of occupational safety, without exceeding costs and without significant risks or harm to the safety of workers on the project site. These are not easy targets, because construction sites are complex and crowded, and because of the constant pressures of time and constant changes in the work environment [27,28]. An accident at work is an unexpected event caused by an external reason, resulting in injury or death, which takes place in connection with work. In general, lack of knowledge, training, poor supervision, and the lack of safe means to carry out activities are the main causes of work accidents. In addition to negligence, complete recklessness, and indifference, the construction industry is inherently short-term, complex, and diverse, and lacks internal and external oversight. All these factors have an impact on the performance of occupational safety within the workplace [29]. In construction, unsafe behavior is the most important factor in the occurrence of construction accidents, which leads directly from the fact that a safety culture does not exist on the worksite [30]. Accidents are associated with a variety of reasons, and deviation from the essence of the work may be the main cause [31]. Masayuki [32] recognized one of the methods for inculcating a safety culture in the workplace is by giving awards to the workers. Active communication and information transfer between management and workers have been suggested to produce better safety standards, and improve safety policies [33]. Discussing occupational safety in frequent meetings raises the level of safety in planning, which leads to the significant development of occupational safety; while reviewing occupational safety during progress meetings makes predicting problems easy and specific [34]. Wong and Gray [35], Tam and Zeng [36], and Ghani and Baki [37] listed various responsibilities and specifications that safety engineers should undertake to ensure the occupational safety of a project, such as: the number of safety engineers should be appropriate to the volume of project, and the safety engineer should have powers to stop all events in the projects in case of hazards happening. 

There are four categories of barrier affecting safety performance in infrastructure projects that should be investigated; these are: (1) management, (2) safety culture, (3) behavior, and (4) awareness [38]. Safety performance can be influenced by factors including the top management’s and project managers’ poor safety awareness, poor safety resources, absence of training, and irresponsible operations [36]. Safety management can be regarded as the totality of practices implemented by the management of an organization to control hazards, so that occupational accidents and injuries are ultimately prevented, i.e., different strategies and activities focusing on the safety of the workers [39]. Safety management, thus, involves a wide range of both formal and informal practices, focusing on standardized activities, such as risk assessment on the one hand, and managers actively showing a commitment to safety on the other [40]. Safety management relates to the actual practices, roles, and functions associated with remaining safe [41]. The application of effective occupational safety management in the workplace makes the existing system safer and less susceptible to accidents and injuries. Most of the research conducted on occupational safety systems has shown a clear increase in injury rates due to insufficient occupational safety regulations in the workplace [42,43].

The contractor should perform an internal audit of their safety management program, and take advantage of past safety experience to periodically update safety plans [44]. The selection of contractors bidding for construction projects should be based on occupational safety considerations, including an assessment of the contractor’s previous occupational safety performance, current programs and practices, and the contractor’s attitude towards safety [45,46]. The safety management needs to be aware of an accident and record it, to establish a better course of action for future accident or incident prevention. If the accident is not reported or detailed, it is impossible to identify the causes [47].

The negligence of workers is a cause of workplace accidents, but is not the only reason, as failure to control occupational safety within the project is a shared responsibility with management. It is assumed that occupational safety systems have a financial allocation to the fullest extent possible, and that the management is fully aware of the cost involved, measure the level of accidents, and improve the safety performance of the project, through reduced risks and the number of injuries [48].

In Kuwait, construction company management believed that occupational safety procedures and requirements increase the cost of construction. However, it was also found that the costs of accidents and safety measures were not considered in the contractor’s offer, and only the cost of insurance of the project was considered, among all elements of occupational safety [49]. Improving the quality of occupational safety needs the adoption of an effective and complete safety approach at the management level; this approach should be clear, through careful consideration of each serious activity identified at the design stage or subsequent stages, in addition to holding training programs on occupational safety and health for old and new workers, where the training is based on the correct decisions in dealing with accidents associated with their workplace [50]. Infrastructure and construction risks generally coincide with many things, but infrastructure is affected by many restrictions, requirements, and safety aspects that are generally not fully controlled, such as in terms of infrastructure, or occupational safety related to the care of “third party” public security. These restrictions may be related to political or social matters [51]. Knowing and evaluating past incidents and preparing their records is critical for management that wants to minimize future incidents in their company. Writing reports, directing them to the competent department, and reviewing them to take the correct action has a significant role in minimizing accidents, in addition to a follow-up program for evaluating the effectiveness of the procedures. 

Safety climate is a summary concept describing the safety ethic in a company or workplace which is reflected in workers “beliefs about safety and is supposed to forecast the way workers behave concerning safety in that workplace” [50]. The study conducted by Dedobbeleer and Béland [52] on construction workers, suggested that the management’s commitment to safety, and workers’ involvement in safety were the two most important factors impacting the safety climate. In a study on road workers, two separate sets of principles were found for workers and supervisors, both of which include attitudes towards occupational safety, and supplied variables in labor requirements and occupational safety as part of the work [53]. 

Important requirements include alerting new workers to potential hazards during their work and making an effort to comply with safety regulations, including wearing personal protective equipment (such as safety glasses and hard hats), placing barricades, and deploying warning signs around drilling places [54]. The training of administrative staff, with appropriate awareness and competence for the requirements of occupational safety, leads to obtaining an effective program and plan for the performance of occupational safety and its implementation, especially to senior management because they are drawing the outlines of a positive occupational safety culture, and applying it to the work and workers as required [55].

Implementation of occupational safety requirements on the worksite needs a true safety culture and appropriate feedback from workers. Activating occupational safety training and guidance programs is essential to improving the workers’ safety culture and attitudes towards safety, in addition to educating workers about safety rules and procedures, and training workers on safe working methods. Well-trained workers are allowed to carry out dangerous and important work. Workers’ participation in occupational safety programs should be mandatory, as they are exposed to daily work hazards [56]. 

Pratt and Fosbroke [57] emphasized that the best technique to develop occupational safety in the workplace is to change the behavior and performance of individuals with the requirements of occupational safety. The development of occupational safety is reactive and behavior-based, and is an effective tool for increasing occupational safety in the workplace; this includes allowing workers to monitor each other to support the development of safety habits through discussion and analysis tools. This method might be considered to have a limited impact, due to the many variations that occur daily in the workplace, so that the proposed improvements are not feasible, this therefore may require the increased behavioral awareness of supervisors, to be more aware of existing physical risks [58].

A main cause of accidents is the poor safety performance of the workers, as one from a combination of contributing causes [38]. Moreover, in recent years safety performance has increasingly come to center around the safety-related behaviors of workers [59]. Occupational safety behaviors give a clear indication of the presence or absence of occupational safety in the workplace [60]. These behaviors, in turn, are often located in a broader social and organizational context, where a multitude of conditions affect safety outcomes [61].

Difficulties in communication with foreign workers can also cause accidents on the construction site [62]. The safety issue faced is due to some of the workers not speaking or understanding the local language. Effective communication might help both parties, namely the safety supervisor and the worker, to have better coordination at the construction site [37]. Good communication affects the implementation of the safety program; when the management communicates openly with the worker, the worker will report on unsafe working practices and hazardous environments [63].

Unsafe behaviors are often the source of many accidents, together with poor working conditions, malfunctioning equipment, poor systems, and poorly designed processes; all of which can encourage unsafe behaviors. While, a company’s attitude towards safe work is an important factor affecting the implementation of occupational safety requirements. It is not enough to provide safe equipment, systems, and procedures if the culture does not promote safe and healthy work [64]. A vast majority of the studies focusing on safety climate and safety culture thus have pointed towards the importance of safety management, including a leadership style that promotes trust and open communication, and makes safety a prioritized area [65]. From the literature review carried out, various barriers were highlighted and modified, as shown in Table 1, as per the working environment of the Gaza Strip.

## 3. Methodology

In this study, descriptive, analytical, and cross-sectional design were utilized, as these are the appropriate design techniques to describe relationships in the collected data [76]. Qualitative and quantitative survey methods were implemented to check the set hypothesis to evaluate the barriers in infrastructure projects in the Gaza Strip. For study purpose, five governorates were selected within the Gaza Strip, i.e., the KhanYounis governorate, Northern governorate, Middle governorate, Rafah governorate, and Gaza governorate. The methodology of this research consisted of six main steps, as shown in Figure 2. 

### 3.1. Questionnaire Design and Development

Initially, factors were identified from the literature review stage, and then modified through a pilot survey. In the questionnaire design stage, validity and reliability were reviewed to form a final questionnaire. The questionnaire was presented to the Institutional Review Committee (IRC), and the details of the committee are provided in Table A1, Appendix A. The IRC checked the questionnaire fulfilled the code of ethics, and that no sensitive questions were involved. Final IRC approval was given after a check of the questionnaire’s suitability to the study scope, and all the questionnaire’s questions were required to achieve the study objectives.

### 3.2. Study Population and Sample Size

Consultants and contractors who are involved in infrastructure projects in the Gaza Strip were selected as the only respondents for this study, where a random selected among the population was made. Due to the large population size, Equation (1) was utilized to establish the sample size [77]:(1)$$SS=\frac{Z^{2}\times P\times \left(1-P\right)}{C^{2}}$$
where “*SS*” is the sample size, “*Z*” is the *Z* value (i.e., 1.96 at 95% confidence level), “P” is the percentage picking a choice (0.5 taken in this case), and “*C*” is the confidence interval (0.05 ± 5). Using Equation (1), the sample size comes to 384, whereas, for correction for a finite population, Equation (2) was utilized:(2)$$New\;SS=\frac{\mathrm{ss}}{1+\frac{ss-1}{population}}$$

Using Equation (2), the sample size came to 53 for 62 consultants, based on a 95% confidence level. A distribution of 53 questionnaires was made among consultants, of which only 36 gave feedback, a 67.92% response rate.

In the same manner as for consultants, the sample size was calculated by Equation (2) for 122 contractors, and the size came to 96, at a 95% confidence level. The distribution was made among the 96 contractors, and 100% feedback was received. The questionnaire was developed based on the literature review, and was modified through interviews with field experts. Their suggestions were carefully taken into consideration, and based on which, modifications were made before making the distribution. Figure 3 shows the study distribution among the population. The 132 responses were gathered from the Gaza Strip and were statistically analyzed, based on which conclusions were made.

### 3.3. Validity and Reliability of Study Instruments

To check that the content of the questionnaire was valid, face validity was performed, where field experts judged the questionnaire content. For checking the consistency of the data, reliability analysis was performed. By these means, much useful information came out, which was incorporated in the developed questionnaire. Statistical Package for Social Science (SPSS-V22) was used to perform the following data analyses on the gathered information: frequency, mean, standard deviation, relative importance index, normal distribution, and Pearson’s correlation coefficient.

## 4. Results and Discussion

### 4.1. Validity and Reliability Analysis

Before making the full distribution, it is better to observe the feedback of a limited population, which tells us whether the questionnaire portrays what it is meant for. For this purpose, a pilot survey was made among 30 individuals, i.e., consultants and contractors of the construction industry.

Two statistical tests were applied to ensure questionnaire validity. The first was internal validity, and the second was structural validity. Internal validity was assessed to measure the correlation coefficient between every item. Whereas, structural validity was to examine the correlation between every dimension and whole dimensions which have a similar scale.

Table 2 shows the correlation coefficient and *p*-value for each item of the duties of competent authorities in safety practices in the infrastructure sector, referring to the internal validity. The *p*-values are less than 0.05, so the correlation coefficients of each item are significant at α = 0.05. Thus, it can be said that the items of each item are consistent and valid to measure what they were set for.

As shown in Table 3, the significance values are less than 0.05, thus the fields are structurally valid, and further assessment can be performed.

Reliability is the degree of consistency, and is performed to measure the consistency of the data. A value of 0.7 and above is considered as acceptable. For this purpose, Cronbach’s coefficient alpha was used. For each item, the Cronbach’s coefficient alpha (Cα) was computed, where every value was above 0.7 as shown in Table 4. Thus, the outcome shows the questionnaire reliability.

### 4.2. General Information of Study Participants

The distribution of 132 responses is shown in Figure 4.

It was observed that most of the companies were keeping their safety budget at less than 1% of their project cost, showing negligence by the higher authorities in implementing safety within the construction projects. Though it seems a tactic to save the money, it later results in miserable effects, in terms of project cost or a higher fatality rate.

### 4.3. Research Hypothesis

**Hypothesis.** 
*The barriers affect the implementation of safety practices in the infrastructure sector in the Gaza Strip at a significance level of 0.05.*


#### 4.3.1. Safety Policy Barriers

This part consists of eight items to assess safety policy barriers in improving safety performance in construction projects in the Gaza Strip. From Table 5, it is seen that “A contractor committed to an occupational safety program is not rewarded” was ranked first by both the consultants and contractors, with 0.674 Relative Importance Index (RII). Even with a separate rank by each, this factor stood first, with 0.694 and 0.709 RII. This indicates the importance of motivation in occupational safety programs, and shortcomings in this matter reflected negatively on the development of occupational safety, which is what Patrick [72] pointed out; that legislation and regulation implementation will not reduce the number of accidents unless there is a sufficient budget provided for health and safety management and its implementation.

On the other hand, it is shown that “The contractor does not punish those who are not committed to occupational safety” was placed second by both the consultants and contractors, with 0.683 RII. In a separate rank by each, this factor stood as third, with 0.667 and 0.678 RII. This is identical to Fang and Huang [78], where it was said that to prevent the workers from repeating their offences; they should be penalized. 

“The contractor should not allocate a special budget for occupational safety” was ranked in the third position by both the consultants and contractors, with RII of (0.673). The responding contractors ranked this factor in the second position, with RII of (0.683), while it was ranked in the sixth position by the responding consultants, with RII of (0.650). This indicates the need for a real orientation to make the safety budget a priority in infrastructure projects. Foo [69] concluded that without an adequate budget covering OSH (Occupational Safety and Health) requirements and their implementation, implementing regulations will not be sufficient and will not reduce accidents.

Last, “There is no insurance policy for the project and the workers” was ranked last by both the consultants and contractors, with 0.484 RII. Additionally, each of them separately ranked it in the last position, with RII of (0.511) and (0.484), respectively. This is because insurance is applicable in most infrastructure projects.

For factors in the safety policy barriers group, the correlation coefficient comes to 0.762, with a 0.000 *p*-value (Sig.). As the *p*-value is less than the significance value, i.e., α = 0.05, this indicates that there is a relationship among the contractors and the consultants within this group.

#### 4.3.2. Management Barriers

This part consists of 20 items to assess the management barriers to improving safety performance in construction projects in the Gaza Strip. From Table 6, it is seen that “The safety engineer does not have significant powers, such as stopping work when needed” was placed first by both the consultants and contractors, with 0.718 RII. In addition, in a separate rank by each, this factor again stood first, with 0.672 and 0.737 RII. This relates to the powers given to the safety engineer, his greater role indicates a real interest in occupational safety in the project. Wong and Gray [35] said that the safety engineer should have powers to stop all events in the projects in case of hazards happening.

On the other hand, it is shown that “The number of safety engineers does not match the size of the project” was ranked second by both the consultants and contractors, with 0.708 RII. A separate rank of this factor by the contractors ranked this factor as second, with 0.725 RII, while it was placed fourth by the consultants, with 0.663 RII. This should be referred to in the contractual papers as binding between the parties, depending on the size and quality of the project. This is consistent with what Tam and Zeng [36] explained.

“There is no real partnership between the parties of the project to commit to safety.” was ranked third by both the consultants and contractors, with 0.693 RII. In a separate ranking by contractors, this factor placed third, with 0.710 RII, while it was placed seventh by the consultants, with 0.650 RII. Vinodkumar and Bhasi [39] noted the importance of cooperation in the work environment. This is often caused by the negative atmosphere in the work environment, and a lack of real concern for occupational safety, and considered as a secondary to the costs and speed of delivery.

Last, “The owner does not care about the application of occupational safety program” was ranked last by both the consultants and contractors, with 0.553 RII. In addition, it stood last in a separate ranking by each, with 0.539 and 0.548 RII. This makes sense for the owner to ensure that his work is done without any problems or dangers.

For factors in the management barriers group, the correlation coefficient came to 0.550, with 0.012 *p*-value (Sig.). As the *p*-value is less than significance value, i.e., α = 0.05, this indicates that there was a relationship among the contractors and the consultants within this group.

#### 4.3.3. Behavioral and Cultural Barriers

This part consists of 11 items to assess behavioral and cultural barriers to improving safety performance in construction projects in the Gaza Strip. From Table 7, it is seen that “Workers who are not committed to occupational safety are not excluded” was ranked first by both the consultants and contractors, with 0.718 RII. In addition, in a separate rank by each, this factor stood first, with 0.744 and 0.737 RII. This high percentage demonstrates the importance of these barriers, and it can be said that the lack of stringent laws, and perhaps the operating system in infrastructure projects based on kinship and friendship, adds to this challenge.

On the other hand, it is shown that “The new worker is more exposed to occupational safety accidents” was placed second by both the consultants and contractors, with 0.711 RII. In a separate rank by the contractors, this factor stood third, with 0.708 RII, while it was placed sixth by the consultants, with 0.644 RII. Inexperience is dangerous, because dealing with equipment and machinery needs a high level, especially in a specialized matter, such as infrastructure, even with the use of simple tools. Due to this, experts added this factor to the questionnaire.

“Government agencies do not give adequate instructions to contractors and workers about safety regulations” was ranked third by both the consultants and contractors, with 0.702 RII. In a separate rank by the contractors, this factor stood second, with 0.723 RII, while it was placed fifth by the consultants, with 0.650 RII. This drives the government to pay more attention and improve safety factors, and enact appropriate laws to improve the work environment.

Last, “Language and culture are a barrier in strengthening occupational safety within the project” was placed last by both consultants and contractors, with RII. In addition, in a separate rank by each, this factor stood last, with 0.578 and 0.607 RII. Difficulties in communication with foreign workers can also cause accidents at the construction site. However, not including this item as a barrier is of great importance because the environment of the Gaza Strip is in one language and one culture.

For factors in behavior and culture barriers group, the correlation coefficient came to 0.598, with 0.002 *p*-value (Sig.). As the *p*-value was less than the significance value, i.e., α = 0.05, this indicates that there is a relationship among the contractors and the consultants within this group.

## 5. Conclusions

In this research, barriers to implementing occupational safety in infrastructure projects were evaluated. A questionnaire comprised of 39 barriers was made to get feedback from consultants and contractors in the Gaza Strip. After performing an analysis via SPSS, the results revealed that in “Safety policy barriers” both consultants and the contractors ranked “A contractor committed to an occupational safety program is not rewarded” as first. In “Management barriers”, the item “The safety engineer does not have significant powers, such as stopping work when needed” was ranked first. While in “Behavioral and cultural barriers”, the item “Workers who are not committed to occupational safety are not excluded” was ranked first by both the consultants and the contractors, showing the importance of these barriers in the infrastructure projects, and that negligence can cause serious threats to the project. Looking at the responses made by both consultants and contractors, a strong convergence of views can be seen, which gives a clear indication of the extent to which the problems they face are shared. A single environment and the forced closure of the Gaza Strip may be one of the reasons for this convergence, as well as the relative tradition among all construction work institutions in public policies, plans, and procedures. Occupational safety cannot be determined in infrastructure projects without the elimination of the existing barriers. This study sets a benchmark for construction stakeholders, especially policymakers, to observe the influential factors causing distress, and to make the changes in their planning, accordingly. The outcome of this study is also useful for countries having a similar geography. 

## 6. Future Direction

It is recommended to deeply investigate occupational safety during other phases (design, operation, and demolition) in infrastructure projects in the Gaza Strip. The relationship between project parties (owner, consultant, contractor, worker) separately for each, and their impact on the application of occupational safety can be evaluated. The study can further be extended in field projects by developing a checklist of all the barriers which need to be cleared on-site. Moreover, a comparison of these barriers can be made with the existing health and safety guidelines in Palestine, and any lack must be eliminated to support the occupational safety implementation in infrastructure projects. Once the barriers are fully addressed then this research could be further extended, and a correlation of the existing issues could be made with developing countries’ construction industries.

## Figures and Tables

**Figure 1 ijerph-18-03553-f001:**
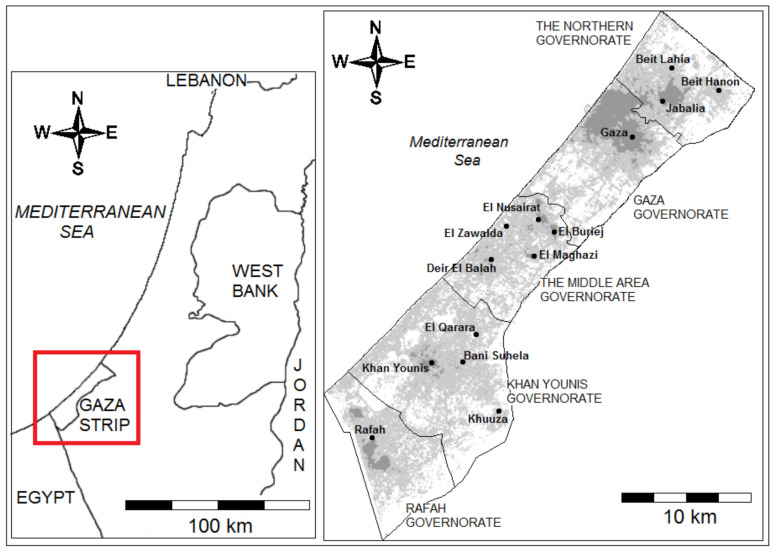
Location map of the Gaza Strip [24].

**Figure 2 ijerph-18-03553-f002:**
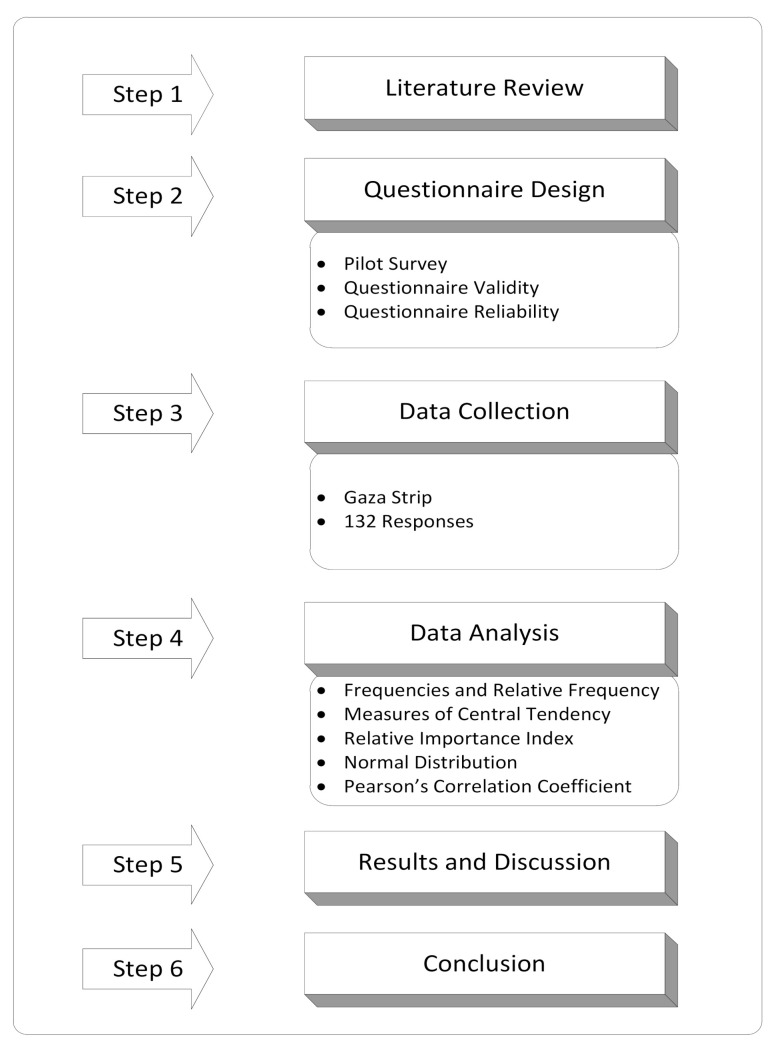
Research Flowchart.

**Figure 3 ijerph-18-03553-f003:**
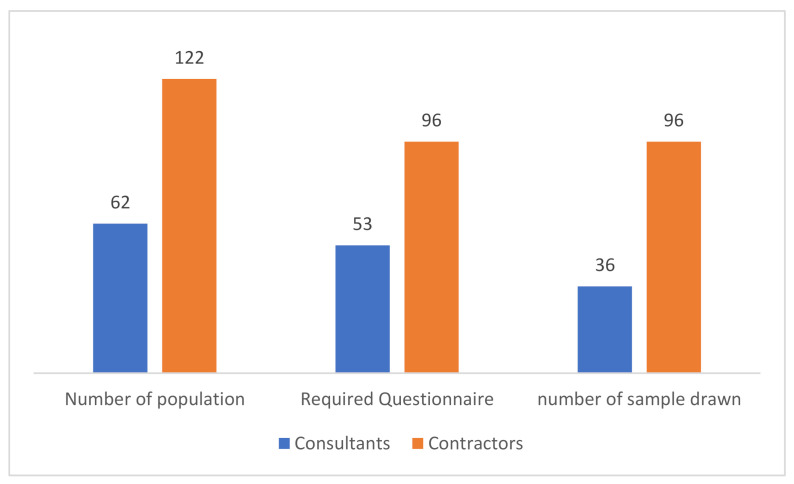
Population size and study sample.

**Figure 4 ijerph-18-03553-f004:**
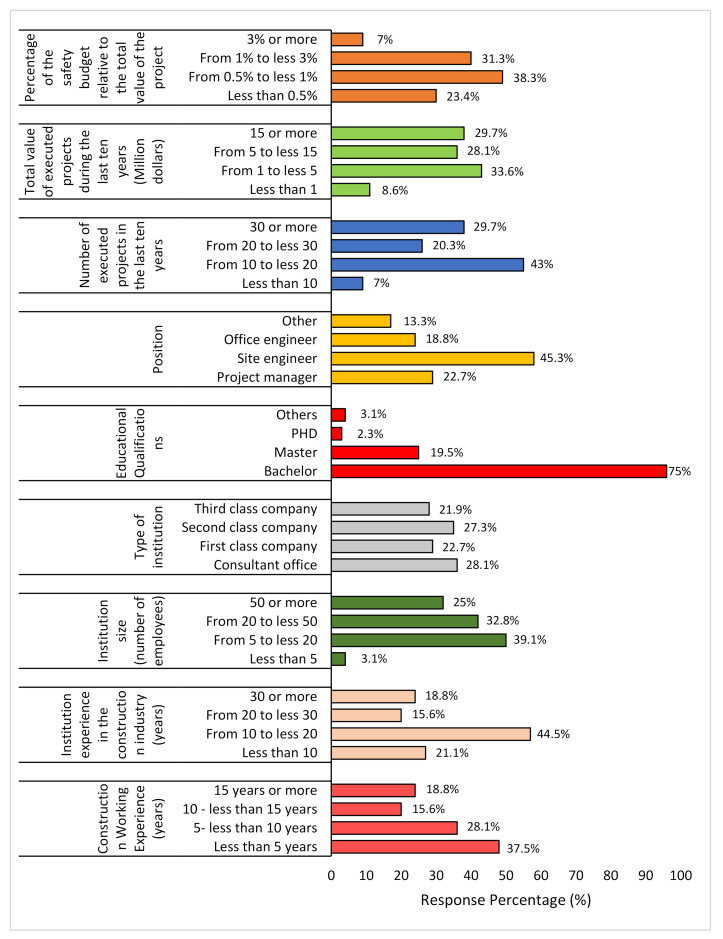
Distribution of study participants.

**Table 1 ijerph-18-03553-t001:** List of barriers that hinder the implementation of safety practices in the infrastructure sector.

No	Identified Barriers from the Literature Review	Comment	Reference
**Safety policy barriers**			
1	The contractor does not have a clearly stated occupational safety policy	Selected	[66,67]
2	Weak contractor implementation of occupational safety policy	Selected	[68]
3	The contractor should not allocate a special budget for occupational safety	Selected	[69]
4	The contractor does not punish those who are not committed to occupational safety	Selected	[70,71]
5	A contractor committed to an occupational safety program is not rewarded	Selected	[72]
6	Staff ignorance of the occupational safety policy	Selected	[67]
7	There is no insurance policy for the project and the workers	Selected	[73]
8	Work injuries are not documented within the project	Selected	[47]
**Management barriers**			
9	The owner does not care about the application of occupational safety program.	Selected	[50,68]
10	The contractor does not care about the application of the occupational safety program.	Selected	[50,74]
11	The consultant is not interested in implementing the occupational safety program.	Selected	[50,68]
12	Government agencies do not monitor the occupational safety program in projects.	Selected	[50,68]
13	There is no safety engineer in the project.	Selected	[29]
14	Safety rules are not respected.	Selected	[29]
15	There are no regular Occupational Safety and Health (OSH) meetings within the project.	Selected	[34]
16	No occupational safety meetings are held within the project.	Selected	[34]
17	Those who are committed to an occupational safety program are not motivated.	Selected	[32]
18	Poor coordination between different project teams.	Selected	[33]
19	The number of safety engineers does not match the size of the project.	Selected	[35,36,37]
20	The safety engineer does not have significant powers, such as stopping work when needed.	Selected	[35,36,37]
21	There is no real partnership between the parties of the project to commit to safety.	Selected	[39]
22	Safety plans are not updated after the end of each project.	Selected	[44,47]
23	The safety procedures do not include the third party (public).	Selected	[51]
24	The contractor’s occupational safety history is not important in awarding the tender.	Selected	[45]
25	Tenders do not include mandatory safety requirements.	Selected	[49]
26	No safety courses are given before the start of the project.	added	-
27	Staff do not receive training courses to deal with equipment and machinery within the project.	added	-
28	Modern technology is not used to provide safety.	added	-
**Behavioral and cultural barriers**			
29	Project staff lack full awareness of occupational safety.	Selected	[47,55,75]
30	Staff do not meet occupational safety instructions.	Selected	[71]
31	The staff do not have a thorough knowledge of the use of machinery and equipment.	Selected	[54,56]
32	Staff do not have experience in dealing with emergencies.	Selected	[54]
33	Employees accept working in an environment that does not respect safety standards.	Selected	[64]
34	Government agencies do not give adequate instructions to contractors and workers about safety regulations.	added	-
35	Work pressure and productivity ensure reduced commitment to occupational safety.	Selected	[54]
36	Use of tools and machinery that are dangerous to workers.	Selected	[64]
37	Language and culture are a barrier in strengthening occupational safety within the project.	added	-
38	The new worker is more exposed to occupational safety accidents.	added	-
39	Workers who are not committed to occupational safety are not excluded.	added	-

**Table 2 ijerph-18-03553-t002:** Internal validity of barriers that hinder the implementation of safety practices in the infrastructure sector.

No.	Fields	Correlation Coefficient	*p*-Value
**Safety policy barriers**			
1	The contractor does not have a clearly stated occupational safety policy.	0.684	0.000 *
2	Weak contractor implementation of occupational safety policy.	0.694	0.000 *
3	The contractor should not allocate a special budget for occupational safety.	0.652	0.000 *
4	The contractor does not punish those who are not committed to occupational safety.	0.665	0.000 *
5	A contractor committed to an occupational safety program is not rewarded.	0.623	0.000 *
6	Staff ignorance of the occupational safety policy.	0.685	0.000 *
7	There is no insurance policy for the project and the workers.	0.513	0.000 *
8	Work injuries are not documented within the project.	0.654	0.000 *
**Management barriers**			
9	The owner does not care about the application of the occupational safety program.	0.527	0.000 *
10	The contractor does not care about the application of the occupational safety program.	0.530	0.000 *
11	The consultant is not interested in implementing the occupational safety program.	0.490	0.000 *
12	Government agencies do not monitor the occupational safety program in projects.	0.564	0.000 *
13	There is no safety engineer in the project.	0.556	0.000 *
14	Safety rules are not respected.	0.631	0.000 *
15	There are no regular OSH meetings within the project.	0.606	0.000 *
16	No occupational safety meetings are held within the project.	0.589	0.000 *
17	Those who are committed to an occupational safety program are not motivated.	0.536	0.000 *
18	Poor coordination between different project teams.	0.560	0.000 *
19	The number of safety engineers does not match the size of the project.	0.612	0.000 *
20	The safety engineer does not have significant powers, such as stopping work when needed.	0.507	0.000 *
21	There is no real partnership between the parties of the project to commit to safety.	0.660	0.000 *
22	Safety plans are not updated after the end of each project.	0.654	0.000 *
23	The safety procedures do not include the third party (public).	0.694	0.000 *
24	The contractor’s occupational safety history is not important in awarding the tender	0.513	0.000 *
25	Tenders do not include mandatory safety requirements.	0.664	0.000 *
26	No safety courses are given before the start of the project.	0.609	0.000 *
27	Staff do not receive training courses to deal with equipment and machinery within the project.	0.598	0.000 *
28	Modern technology is not used to provide safety.	0.565	0.000 *
**Behavior and culture barriers**			
29	Project staff lack full awareness of occupational safety.	0.655	0.000 *
30	Staff do not meet occupational safety instructions.	0.697	0.000 *
31	The staff do not have a thorough knowledge of the use of machinery and equipment.	0.644	0.000 *
32	Staff do not have experience in dealing with emergencies.	0.710	0.000 *
33	Employees accept working in an environment that does not respect safety standards.	0.667	0.000 *
34	Government agencies do not give adequate instructions to contractors and workers about safety regulations.	0.632	0.000 *
35	Work pressure and productivity ensure reduced commitment to occupational safety.	0.689	0.000 *
36	Use of tools and machinery that are dangerous to workers.	0.672	0.000 *
37	Language and culture are a barrier in strengthening occupational safety within the project.	0.617	0.000 *
38	The new worker is more exposed to occupational safety accidents.	0.633	0.000 *
39	Workers who are not committed to occupational safety are not excluded.	0.513	0.000 *

* the significance values of *p*-Value are less than 0.05.

**Table 3 ijerph-18-03553-t003:** Structure validity of the questionnaire.

Dimensions	Correlation Coefficient	*p*-Value
**Barriers that hinder the implementation of safety practices in the infrastructure sector**		
Safety policy barriers	0.657	0.000 *
Management barriers	0.917	0.000 *
Behavioral and cultural barriers	0.867	0.000 *

* the significance values of *p*-Value are less than 0.05.

**Table 4 ijerph-18-03553-t004:** Cronbach’s coefficient alpha for reliability (Cα).

Dimensions	Cronbach’s Alpha (Cα)	*p*-Value
**Barriers that hinder the implementation of safety practices in the infrastructure sector**		
Safety policy barriers	0.768	0.000 *
Management barriers	0.870	0.000 *
Behavioral and cultural barriers	0.859	0.000 *
**Total**	**0.911**	**0.000 ***

* the significance values of *p*-Value are less than 0.05.

**Table 5 ijerph-18-03553-t005:** Rank and Relative Importance Index (RII) of items related to safety policy barriers.

No	Factors	Both		Consultant		Contractor	
		RII	Rank	RII	Rank	RII	Rank
1	The contractor does not have a clearly stated occupational safety policy.	0.638	6	0.661	5	0.628	6
2	Weak contractor implementation of occupational safety policy.	0.664	4	0.667	4	0.663	4
3	The contractor should not allocate a special budget for occupational safety.	0.673	3	0.650	6	0.683	2
4	The contractor does not punish those who are not committed to occupational safety.	0.683	2	0.694	1	0.678	3
5	A contractor committed to an occupational safety program is not rewarded.	0.697	1	0.667	3	0.709	1
6	Staff ignorance of the occupational safety policy.	0.647	5	0.678	2	0.635	5
7	There is no insurance policy for the project and the workers.	0.484	8	0.511	8	0.474	8
8	Work injuries are not documented within the project.	0.496	7	0.554	7	0.476	7
	All items	0.623		0.635		0.618	

**Table 6 ijerph-18-03553-t006:** Rank and RII of items related to management barriers.

No	Factors	Both		Consultant		Contractor	
		RII	Rank	RII	Rank	RII	Rank
1	The owner does not care about the application of the occupational safety program.	0.553	20	0.539	20	0.548	20
2	The contractor does not care about the application of the occupational safety program.	0.614	18	0.644	11	0.602	18
3	The consultant is not interested in implementing the occupational safety program.	0.556	19	0.578	18	0.559	19
4	Government agencies do not monitor the occupational safety program in projects.	0.655	12	0.661	5	0.653	15
5	There is no safety engineer in the project.	0.663	11	0.644	12	0.670	11
6	Safety rules are not respected.	0.671	9	0.628	15	0.688	7
7	There are no regular OSH meetings within the project.	0.673	7	0.633	13	0.689	6
8	No occupational safety meetings are held within the project.	0.667	10	0.617	16	0.687	8
9	Those who are committed to an occupational safety program are not motivated.	0.672	8	0.661	6	0.677	10
10	Poor coordination between different project teams.	0.654	13	0.611	17	0.670	12
11	The number of safety engineers does not match the size of the project.	0.708	2	0.663	4	0.725	2
12	The safety engineer does not have significant powers, such as stopping work when needed.	0.718	1	0.669	2	0.737	1
13	There is no real partnership between the parties of the project to commit to safety.	0.693	3	0.650	7	0.710	3
14	Safety plans are not updated after the end of each project.	0.689	4	0.667	3	0.698	4
15	The safety procedures do not include the third party (public).	0.651	15	0.629	14	0.659	14
16	The contractor’s occupational safety history is not important in awarding the tender.	0.680	5	0.650	8	0.691	5
17	Tenders do not include mandatory safety requirements.	0.636	16	0.561	19	0.665	13
18	No safety courses are given before the start of the project.	0.678	6	0.672	1	0.680	9
19	Staff do not receive training courses to deal with equipment and machinery within the project.	0.652	14	0.650	9	0.653	16
20	Modern technology is not used to provide safety.	0.623	17	0.650	10	0.612	17
	All items	0.655		0.634		0.664	

**Table 7 ijerph-18-03553-t007:** Rank and RII of items related to behavioral and cultural barriers.

No	Factors	Both		Consultant		Contractor	
		RII	Rank	RII	Rank	RII	Rank
1	Project staff lack full awareness of occupational safety.	0.647	8	0.622	9	0.657	7
2	Staff do not meet occupational safety instructions.	0.652	7	0.667	2	0.646	8
3	The staff do not have a thorough knowledge of the use of machinery and equipment.	0.616	10	0.633	7	0.609	10
4	Staff do not have experience in dealing with emergencies.	0.666	6	0.650	4	0.672	6
5	Employees accept working in an environment that does not respect safety standards.	0.683	4	0.653	3	0.693	5
6	Government agencies do not give adequate instructions to contractors and workers about safety regulations	0.702	3	0.650	5	0.723	2
7	Work pressure and productivity ensure reduced commitment to occupational safety.	0.679	5	0.628	8	0.699	4
8	Use of tools and machinery that are dangerous to workers.	0.617	9	0.606	10	0.622	9
9	Language and culture are a barrier in strengthening occupational safety within the project.	0.598	11	0.578	11	0.607	11
10	The new worker is more exposed to occupational safety accidents.	0.711	2	0.644	6	0.708	3
11	Workers who are not committed to occupational safety are not excluded.	0.718	1	0.744	1	0.737	1
	All items	0.663		0.643		0.670	

## Data Availability

The data has been provided in this paper.

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
