# Peer review of "Barriers of Occupational Safety Implementation in Infrastructure Projects: Gaza Strip Case"

_ijerph, 2021, doi:10.3390/ijerph18073553_

Round 1
Reviewer 1 Report
The article is interesting. I think is an innovation this kind of approaches and even in the context in which it is developed.
Some comments:
a. Introduction is ok.
b. State-of-the-art is adequate to the case study and the paper description.
c. Table 1 is very descriptive.
d. Good research flowchart (Fig. 1).
e. In methodology, Why authors used a corrected sample size "SS"?. Please, explain.
f. How the sample size were defined? Is it representative? Why?
g. The results could be correlated with other regions around Middle East? or close to Southeastern Europe?
h. Conclusions are really poor. This section must be improved in the revised version of the manuscript.
i. What about future research works? Future research must be included after revision.
j. References are ok. Please, check if all references are cited in the main body.
k. Could be interesting to include a Figure concerning the location context of the analysis?
l. English language and style are fine/minor spell check required.
Good job!.
After revision, I would like to read the paper one more time, mainly results, discussion, conclusions and future research works.
Author Response
|
S. No |
Reviewer Comments |
Corrections Made |
Remarks |
|
1 |
Introduction is ok. |
Thank you for your appreciation. |
N/A |
|
2 |
State-of-the-art is adequate to the case study and the paper description. |
Thank you for your appreciation. |
N/A |
|
3 |
Table 1 is very descriptive. |
Thank you for your appreciation. |
N/A |
|
4 |
Good research flowchart (Fig. 1). |
Thank you for your appreciation. |
N/A |
|
5 |
In methodology, Why authors used a corrected sample size "SS"?. Please, explain. |
Corrected sample size shows that how much distribution we need to make when the population is finite. Therefore, first “SS” was calculated which gave us an established sample size and then correction was made for consultants and contractors population. |
N/A |
|
6 |
How the sample size were defined? Is it representative? Why? |
The sample size was defined by Equation 2 as the population of consultant and contractor was known. Afterwards, the distribution was made randomly. |
N/A |
|
7 |
The results could be correlated with other regions around Middle East? or close to Southeastern Europe? |
There are 2 main reasons that the results are not correlated with other regions: 1) The factors were adapted as per the environment of Palestine. 2) The construction industry of Palestine is different from other region industries, where Palestine is way more backward at the moment. The comparison can only be possible if the industries are at the same level. |
N/A |
|
8 |
Conclusions are really poor. This section must be improved in the revised version of the manuscript. |
The conclusion has been revised. |
Refer to the revised manuscript. |
|
9 |
What about future research works? Future research must be included after revision. |
Future direction has been provided. |
Refer to Section 6: Future direction of the revised manuscript. |
|
10 |
References are ok. Please, check if all references are cited in the main body. |
All the references are already been provided in the manuscript. |
N/A |
|
11 |
Could be interesting to include a Figure concerning the location context of the analysis? |
A Figure has been introduced concerning the location context of the analysis. |
Refer to Figure 1 of the revised manuscript. |
|
12 |
English language and style are fine/minor spell check required. |
A thorough grammar and English language check has been performed. |
Refer to the revised manuscript. |
|
13 |
Good job!. After revision, I would like to read the paper one more time, mainly results, discussion, conclusions and future research works. |
Thank you for your appreciation. The manuscript has been revised as per the given suggestion. |
Refer to the revised manuscript. |

Reviewer 2 Report
Dear Authors,
Thank you for your work and for the opportunity to read it. I believe that all studies that focus on safety in the workplace are important. Safety and health at work are important and this type of article allows the awareness of social agents to improve the rules of prevention of occupational risks, for example. For all these reasons, I consider that the article is linked to the objectives of the International Journal of Environmental Research and Public Health.
However, there are certain aspects of this work that would be important to complete, modify and/or expand.
- About the title. Your article is focused on a specific study of the infrastructure construction sector in Gaza Strip. Everything in your article is about Gaza Strip, so it should be included in the title.
- Introduction and Literature review. The problem described in your article is very common in less developed and developing countries. In all these countries there is a regulatory gap in almost all productive activities, including the construction of infrastructure. In this sense, I think it would be interesting to incorporate international bibliography that would make your work even more interesting. I suggest that you study the opportunity to incorporate the following bibliography:
- Kheni, N. A., Gibb, A. G. F., & Dainty, A. R. J. (2010). Health and safety management within small- and medium-sized enterprises (SMEs) in developing countries: Study of contextual influences. Journal of Construction Engineering and Management, 136(10), 1104-1115. doi:10.1061/(ASCE)CO.1943-7862.0000218
- Azhar, S., & Choudhry, R. M. (2016). Capacity building in construction health and safety research, education, and practice in pakistan. Built Environment Project and Asset Management, 6(1), 92-105. doi:10.1108/BEPAM-09-2014-0044
- Kumie, A., Amera, T., Berhane, K., Samet, J., Hundal, N., Michael, F. G., & Gilliland, F. (2016). Occupational health and safety in ethiopia: A review of situational analysis and needs assessment. Ethiopian Journal of Health Development, 30(SpecialIssue1), 17-27.
- Partanen, T. J., Hogstedt, C., Ahasan, R., Aragón, A., Arroyave, M. E., Jeyaratnam, J., . . . Wesseling, C. (1999). Collaboration between developing and developed countries and between developing countries in occupational health research and surveillance. Scandinavian Journal of Work, Environment and Health, 25(3), 296-300. doi:10.5271/sjweh.438
- Sorensen, G., Nagler, E. M., Pawar, P., Gupta, P. C., Pednekar, M. S., & Wagner, G. R. (2017). Lost in translation: The challenge of adapting integrated approaches for worker health and safety for low-and middle-income countries. PLoS ONE, 12(8) doi:10.1371/journal.pone.0182607
- Ugwu, O. O., & Haupt, T. C. (2005). Key performance indicators for infrastructure sustainability – a comparative study between hong kong and south africa. Journal of Engineering, Design and Technology, 3(1), 30-43. doi:10.1108/17260530510815321
The reference Nawi et al (2017) appears in line 103, as reference number 30. In this article they talk about 6 key factors to evaluate the safety climate in construction. You identify four categories of barriers (line 102): 1) management, 2) safety culture, 3) behaviour, and 4) awareness. Later, in section 2.1.2 you explain the last three factors and table 1 reflects three groups of barriers: "Safety policy barriers", "Management barriers", "Behaviour & culture barriers". Could you group the factors into 2, 3 or 4? I think you should better explain the categories into which you divide the barriers.
Also, I believe that your introduction should contain the research question, the objective of this work and the hypothesis you propose to validate. It is not clearly exposed.
3. In the methodological section, I think some aspects need to be improved. Specifically, it is indicated that the sample has been comprised of consultants and contractors. They must define what type of consultants have been interviewed.
Also, the random selection of the sample may not be ideal when the population is heterogeneous. It would be interesting to have a table that shows the size of each enterprise. In many cases, the problems of implementing occupational health and safety regulations are strongly related to the size of the enterprise.
Is the information about the sample in figure 3? It is not clear.
I believe that the methodological section contains items that should be reordered in the results section. I ask you to review its content.
Pilot Study. You say that you have conducted 30 individual surveys of people in the construction industry, consultants or contractors? (lines 251-254).
Internal validity. Table 2. It is very relevant that all the safety indicators have been significant. It is a very important result, can you relate it to similar studies?
Some errors:
- Line 226. Where it says "S" I think it should say "SS".
- Line 229. They should indicate what "pop" is in equation 2.
Finally, I think this study is very interesting and needs to be clarified before being published. I hope that the authors have enough courage to face the work of modification of this article.
Good luck.
Author Response
|
S. No |
Reviewer Comments |
Corrections Made |
Remarks |
|
1 |
About the title. Your article is focused on a specific study of the infrastructure construction sector in Gaza Strip. Everything in your article is about Gaza Strip, so it should be included in the title. |
The title has been modified as per the given suggestion. |
Refer to the revised manuscript. |
|
2 |
Introduction and Literature review. The problem described in your article is very common in less developed and developing countries. In all these countries there is a regulatory gap in almost all productive activities, including the construction of infrastructure. In this sense, I think it would be interesting to incorporate international bibliography that would make your work even more interesting. I suggest that you study the opportunity to incorporate the following bibliography: · Kheni, N. A., Gibb, A. G. F., & Dainty, A. R. J. (2010). Health and safety management within small- and medium-sized enterprises (SMEs) in developing countries: Study of contextual influences. Journal of Construction Engineering and Management, 136(10), 1104-1115. doi:10.1061/(ASCE)CO.1943-7862.0000218 · Azhar, S., & Choudhry, R. M. (2016). Capacity building in construction health and safety research, education, and practice in pakistan. Built Environment Project and Asset Management, 6(1), 92-105. doi:10.1108/BEPAM-09-2014-0044 · Kumie, A., Amera, T., Berhane, K., Samet, J., Hundal, N., Michael, F. G., & Gilliland, F. (2016). Occupational health and safety in ethiopia: A review of situational analysis and needs assessment. Ethiopian Journal of Health Development, 30(SpecialIssue1), 17-27. · Partanen, T. J., Hogstedt, C., Ahasan, R., Aragón, A., Arroyave, M. E., Jeyaratnam, J., . . . Wesseling, C. (1999). Collaboration between developing and developed countries and between developing countries in occupational health research and surveillance. Scandinavian Journal of Work, Environment and Health, 25(3), 296-300. doi:10.5271/sjweh.438 · Sorensen, G., Nagler, E. M., Pawar, P., Gupta, P. C., Pednekar, M. S., & Wagner, G. R. (2017). Lost in translation: The challenge of adapting integrated approaches for worker health and safety for low-and middle-income countries. PLoS ONE, 12(8) doi:10.1371/journal.pone.0182607 · Ugwu, O. O., & Haupt, T. C. (2005). Key performance indicators for infrastructure sustainability – a comparative study between hong kong and south africa. Journal of Engineering, Design and Technology, 3(1), 30-43. doi:10.1108/17260530510815321 |
Thank you for providing the useful articles which have been incorporated within the text. |
Refer to Introduction section of the revised manuscript. |
|
3 |
The reference Nawi et al (2017) appears in line 103, as reference number 30. In this article they talk about 6 key factors to evaluate the safety climate in construction. You identify four categories of barriers (line 102): 1) management, 2) safety culture, 3) behaviour, and 4) awareness. Later, in section 2.1.2 you explain the last three factors and table 1 reflects three groups of barriers: "Safety policy barriers", "Management barriers", "Behaviour & culture barriers". Could you group the factors into 2, 3 or 4? I think you should better explain the categories into which you divide the barriers. |
The literature section has been revised where the numbers discrepancy has been removed. |
Refer to Section 2 of the revised manuscript. |
|
4 |
Also, I believe that your introduction should contain the research question, the objective of this work and the hypothesis you propose to validate. It is not clearly exposed. |
The research question, objective and the hypothesis are mentioned as per the suggestion. |
Refer to the last paragraph of the introduction section. |
|
5 |
In the methodological section, I think some aspects need to be improved. Specifically, it is indicated that the sample has been comprised of consultants and contractors. They must define what type of consultants have been interviewed. |
The methodological section has been revised as per the suggestion. |
Refer to Section 3 of the revised manuscript. |
|
6 |
Also, the random selection of the sample may not be ideal when the population is heterogeneous. It would be interesting to have a table that shows the size of each enterprise. In many cases, the problems of implementing occupational health and safety regulations are strongly related to the size of the enterprise. |
The reason to keep the random selection from the population is due to the fact the occupational health and safety is a problem facing by all the enterprises regardless of their size. These barriers are implying on the overall enterprise of Gaza Strip, not a particular class. |
N/A |
|
7 |
Is the information about the sample in figure 3? It is not clear. |
Figure 3 which is new Figure 4 in the revised manuscript shows the response of percentages received in each category which has been updated in the Figure. |
Refer to Figure 4 in the revised manuscript. |
|
8 |
I believe that the methodological section contains items that should be reordered in the results section. I ask you to review its content. |
Results from the methodological section have been shifted to the results and discussion section. |
Refer to Section 4 of the revised manuscript. |
|
9 |
Pilot Study. You say that you have conducted 30 individual surveys of people in the construction industry, consultants or contractors? (lines 251-254). |
The changes have been made as per the suggestion. |
Refer to Section 4.1 of the revised manuscript. |
|
10 |
Internal validity. Table 2. It is very relevant that all the safety indicators have been significant. It is a very important result, can you relate it to similar studies? |
We did not relate it purposely as the internal validity is of the modified factors, not the one which is exactly adopted. Moreover, not all the similar studies from where the factors have been taken have calculated the internal validity. |
N/A |
|
11 |
Some errors: - Line 226. Where it says "S" I think it should say "SS". - Line 229. They should indicate what "pop" is in equation 2. |
The comment has been addressed. “Pop” is the population, which has been mentioned completed in the equation. |
Refer to Section 3.2 of the revised manuscript. |
|
12 |
Finally, I think this study is very interesting and needs to be clarified before being published. I hope that the authors have enough courage to face the work of modification of this article. Good luck. |
Thank you for such good suggestions which already been incorporated in the revision. |
Refer to the revised manuscript. |

Reviewer 3 Report
This paper evaluated the barriers of implementing the occupational safety in infrastructure projects by issuing questionnaires to consultants and contractors in Gaza Strip. The methodology is sound and well presented. However, there are many places that need to be improved.
My detailed comments are as follows.
- The obstacle to the implementation of occupational safety in infrastructure projects described in this paper is in the Gaza Strip, which should be reflected in the title or abstract.
- The barriers of occupational safety implementation may vary in different regions. It is recommended to add "taking Basha region as an example" to the title.
- Although this research is very meaningful, the theoretical and practical significance of this article should be pointed out at the end of the abstract.
- In the Introduction, the author explains in detail why it is necessary to study the obstacle to the implementation of occupational safety in infrastructure in the Gaza Strip. However, it is insufficient to review whether scholars have studied the obstacle to the implementation of occupational safety in infrastructure in the Gaza Strip.
- The second and third paragraphs of the Introduction spent too much content to explain the difficulties encountered in Pakistan's infrastructure projects, such as the lack of materials. But from the theme of this article, there is no connection between occupational safety and material shortage.
- Since there is no section 2.2 in this article, there is no need to set section 2.1.
- Within the section dedicated to the validity and the reliability of study instruments, a less description could be beneficial. For example, there is no need to assign “Face Validity” as a subsection, it can be integrated into the other sentences. The authors are suggested to refer to Investigating the determinants of contractor’s construction and demolition waste management behavior in Mainland China for the revision of this section.
- The tables, such as tables 2 to 7, are not following the requirements of IJERPH. There are many careless formatting mistakes. In addition, where is Table 4???
- In the Conclusions, there is a lack of the author’s deeper thinking about these barriers, such as what are the future suggestions to solve these barriers.
Author Response
|
S. No |
Reviewer Comments |
Corrections Made |
Remarks |
|
1 |
The obstacle to the implementation of occupational safety in infrastructure projects described in this paper is in the Gaza Strip, which should be reflected in the title or abstract. |
The title has been revised as per the suggestion. |
Refer to the revised manuscript. |
|
2 |
The barriers of occupational safety implementation may vary in different regions. It is recommended to add "taking Basha region as an example" to the title. |
The focus of this study is on the Gaza Strip which is a part of Palestine. We cannot limit this study to a specific region as the outcome is beneficial for all. |
N/A |
|
3 |
Although this research is very meaningful, the theoretical and practical significance of this article should be pointed out at the end of the abstract. |
The abstract of the study has been revised as per the suggestion. |
Refer to Abstract of the revised manuscript. |
|
4 |
In the Introduction, the author explains in detail why it is necessary to study the obstacle to the implementation of occupational safety in infrastructure in the Gaza Strip. However, it is insufficient to review whether scholars have studied the obstacle to the implementation of occupational safety in infrastructure in the Gaza Strip. |
Thorough literature has been performed on Barriers of Occupational Safety Implementation worldwide from where the key obstacles, which are barriers in our case, have been taken and adapted as per the environment of Palestine construction industry after consulting with the field experienced stakeholders. |
N/A |
|
5 |
The second and third paragraphs of the Introduction spent too much content to explain the difficulties encountered in Pakistan's infrastructure projects, such as the lack of materials. But from the theme of this article, there is no connection between occupational safety and material shortage. |
I believe the reviewer has mistakenly mention about Pakistan’s infrastructure projects because the focus of this study is on Palestine infrastructure projects. As far as material shortage linkage to occupational safety is concerned, with the shortage, there is hustle and bustle in the project to complete it on it. By doing that, the focus from the safety is shifted towards the timely completion. In this manner, even materials shortage plays a role in poor occupational safety implementation. |
N/A |
|
6 |
Since there is no section 2.2 in this article, there is no need to set section 2.1. |
Section 2.1 headings and subheadings have been removed. |
Refer to Section 2 of the revised manuscript. |
|
7 |
Within the section dedicated to the validity and the reliability of study instruments, a less description could be beneficial. For example, there is no need to assign “Face Validity” as a subsection, it can be integrated into the other sentences. The authors are suggested to refer to Investigating the determinants of contractor’s construction and demolition waste management behavior in Mainland China for the revision of this section. |
The sections and subsections have been revised as per the given suggestion. |
Refer to Section 3.3: Validity and Reliability of Study Instruments of the revised manuscript. |
|
8 |
The tables, such as tables 2 to 7, are not following the requirements of IJERPH. There are many careless formatting mistakes. In addition, where is Table 4??? |
The formatting of the Tables has been corrected as per the requirement of the journal. Also, Table 4 numbering has been corrected. |
Refer to Table 2 to Table 7 of the revised manuscript. |
|
9 |
In the Conclusions, there is a lack of the author’s deeper thinking about these barriers, such as what are the future suggestions to solve these barriers. |
The conclusion has been revised where the author’s deeper thinking about these barriers has been incorporated along with future suggestions. |
Refer to the conclusion section of the revised manuscript. |

Round 2
Reviewer 1 Report
Improve conclusions and check English.
Author Response
Reply to Reviewer 1
|
S. No |
Reviewer Comments |
Corrections Made |
Remarks |
|
1 |
Improve conclusions and check English. |
The conclusion section has been improved and a proofread of the paper has been made. |
Refer to the revised manuscript. |

Reviewer 2 Report
Dear authors,
Thank you for improving the aspects suggested in my review report. I hope you have had luck with the rest of the review reports.
Regads,
Luis J. Belmonte-Ureña
Author Response
Reply to Reviewer 2
|
S. No |
Reviewer Comments |
Corrections Made |
Remarks |
|
1 |
Dear authors, Thank you for improving the aspects suggested in my review report. I hope you have had luck with the rest of the review reports. |
Thank you for your appreciation. |
N/A |

Reviewer 3 Report
My comments have been addressed.
Author Response
Reply to Reviewer 3
|
S. No |
Reviewer Comments |
Corrections Made |
Remarks |
|
1 |
My comments have been addressed. |
Thank you for your appreciation. |
N/A |
